# The Efficacy of Low-Level Laser Therapy Combined with Single Flap Periodontal Surgery in the Management of Intrabony Periodontal Defects: A Randomized Controlled Trial

**DOI:** 10.3390/healthcare10071301

**Published:** 2022-07-13

**Authors:** S. Silviya, Anitha C.M., P.S.G. Prakash, Sarah Ahmed Bahammam, Maha A. Bahammam, Ammar Almarghlani, Mohammad Assaggaf, Mona Awad Kamil, Sangeetha Subramanian, Thodur Madapusi Balaji, Shankargouda Patil

**Affiliations:** 1Department of Periodontics, SRM Dental College and Hospital, Ramapuram, Chennai 600089, India; drsilviya@yahoo.com (S.S.); anithha@gmail.com (A.C.M.); sangeetha_doc@yahoo.com (S.S.); 2Department of Pediatric Dentistry and Orthodontics, College of Dentistry, Taibah University, Medina 42353, Saudi Arabia; sbahammam@taibahu.edu.sa; 3Department of Periodontology, Faculty of Dentistry, King Abdulaziz University, Jeddah 80209, Saudi Arabia; mbahammam@kau.edu.sa (M.A.B.); ammar.marg@live.com (A.A.); massaggaf@gmail.com (M.A.); 4Executive Presidency of Academic Affairs, Saudi Commission for Health Specialties, Riyadh 11614, Saudi Arabia; 5Department of Preventive Dental Science, College of Dentistry, Jazan University, Jazan 45412, Saudi Arabia; munakamil@yahoo.com; 6Tagore Medical and Dental College, Rathinamangalam, Vandalur, Chennai 600127, India; tmbala81@gmail.com; 7Department of Maxillofacial Surgery and Diagnostic Sciences, Division of Oral Pathology, College of Dentistry, Jazan University, Jazan 45412, Saudi Arabia

**Keywords:** Low-level laser, intra osseous defects, periodontitis, single flap approach

## Abstract

This study aimed at assessing the clinical outcomes of the Single Flap Approach (SFA) with the additional use of Low-level laser therapy (LLLT). The defects were treated as per the principles of SFA, whereby 20 defects received only SFA (control group) and 20 defects received additional LLLT for bio stimulation/bio modulation (test group). Stable primary closure of the flaps was obtained with vertical internal mattress sutures. Plaque indices (FMPS), clinical attachment levels (CAL), probing pocket depth (PPD), and gingival bleeding scores (FMBS) were calculated at baseline, and at the 3rd and 6th months in both groups. An EHI score of 1 was observed at all sites except for two, where a score of 2 in the control group at week 2 was observed. In the test group, the PPD reduction at 6 months was 3.60 ± 0.95 and in the control group it was 3.75 ± 0.91 mm. CAL gain at 6 months was 2.70 ± 1.36 mm and 3.45 ± 1.2 mm in the test group and showed no statistical significance. These data suggested the positive effect of LLLT over CAL gain; thus, LLLT may be combined with SFA to potentially enhance the early wound healing and higher clinical outcomes in terms of increase in CAL and decrease in PPD.

## 1. Introduction

The periodontium is an intricate mosaic of cells and proteins that are primarily responsible for the attachment of teeth. This intricate mosaic bears the brunt of microbial and environmental factors, causing the periodontium to elaborate cytokines that are destructive to the periodontal tissue, leading to an apical migration of the epithelial attachment and loss of periodontal soft and hard tissues.

Various systemic diseases like diabetes, cardiovascular diseases, and vitamin D deficiencies are considered risk factors for periodontal diseases [1,2]. Early attempts to achieve periodontal regeneration included the use of conventional flap procedures, use of barrier membranes, and GTR procedures that accomplish the objectives of epithelial exclusion through controlled cell/tissue repopulation of the periodontal wound, space maintenance, and clot stabilization.

Numerous advancements have been made in our understanding the biology of wound healing in recent years. Wound healing is characterized by proliferation, differentiation, migration, and adhesion of different types of cell. All of these activities are triggered when different growth factors regulate different cell functions, for example, TGFβ and VEGF help in angiogenesis, and connective tissue growth factor and bFGF help in fibroblast and collagen synthesis. These growth factors play a prime role in the healing of periodontal tissue [3].

Moreover, several modulators like Low-Level Laser Therapy (LLLT) have become well-accepted adjuvant medical tools in the healing process of wounds. The main principle of LLLT is based on the bio stimulation effect (Walsh 1997 [4]), which is based on the fact that cellular behavior can be changed by irradiation at a particular wavelength (Hopkins et al., 2004 [5]). This subsequently causes an increase in cell metabolism and proliferation. LLLT is considered to be effective for the treatment of periodontal diseases because of its excellent physical properties, namely, bactericidal effect, cell stimulation, ablation, fibroblast proliferation (Tominaga et al., 1990) [6], hemostasis-enhancing chemotactic activity of leukocytes (Tadakuma et al., 1993 [7]), and proliferation, differentiation and calcification of osteoblastic cells (Yamada T et al., 1991 [8].

It is not only modulators of wound healing that are the primary requisite, however; specific minimally invasive surgical approaches are also important for facilitating wound healing and periodontal regeneration.

Harrel and Ress [9] proposed a minimally invasive surgery in order to achieve minimal flap reflection, minimal wounds, and gentle handling of the periodontal tissues in intrabony defects. Trombelli, in 2007 [10], proposed a single flap approach for reconstructing intrabony defects. This method involves elevation of the limited mucoperiosteal flap only on either the lingual or the buccal side for surgical access so that the adjoining gingival tissues remain intact. This ensures reestablishment of the local vascular supply, adequate wound closure for primary intention, stabilization to the unattached papilla, preservation of gingival aesthetics, and healing.

LLLT is a valid approach in other fields of dentistry such as for the treatment of TMJ disorders [11,12].

Roberto et al. performed the first clinical study on the application of LLLT in sterile pyogranulomatous pododermatitis in dogs, comprising one of the first clinical studies on the use of LLLT in canine dermatology [13].

The rationale for SFA resides the reduction of surgical trauma, increased wound stability, and preservation of the interdental papilla, thereby making it of benefit to reposition the flap and suture placement, thereby optimizing wound closure for primary intention.

Based on the above-mentioned findings, both SFA and LLLT have beneficial effects on probing pocket depth reduction, periodontal wound healing, and obtaining clinical attachment. However, to date, there have been no studies evaluating the adjunctive use of LLLT with SFA in intra osseous periodontal defects and wound healing. Thus, this study is proposed to test two hypotheses, namely, whether the clinically observed outcomes following Single Flap Surgery could be favorably enhanced by the adjunctive use of LLLT, and to observe whether the effect of combining SFA and LLLT could possibly alter and enhance the biology of periodontal wound healing.

## 2. Materials and Methods

### 2.1. Study Design

This was a randomized controlled trial conducted at SRM Dental College, Ramapuram. The workflow of the study is presented in Figure 1. The study proposal was placed before the institutional scientific and ethical review board and was approved before the commencement of the same (Ethical Approval number: SRMU/M&HS/SRMDC/2011/M.D.S-PG Student/507). Written informed consent was obtained. The surgery was performed following the ethical principles of Helsinki. Using SPSS software, the sample size was calculated based on the proportions set at 0.15 with Type II error β set at 90% and type I error α set at 5%. The estimated sample size was 7 sites in each group. To compensate for dropouts, a total of 20 subjects were recruited to each group.

A total of 40 intrabony defects were taken from 40 cases and divided into 2 groups for the study (Group I: 20 cases with SFA (Control); Group II: 20 cases with SFA + LLLT (diode laser) with an intensity of 790–810 nm and with 0.4–0.7 watts (Test)). The inclusion criteria for the study were individuals between 25 and 50 years of age with isolated periodontal pockets >5 mm in depth with a loss of attachment >3 mm. The periodontal pockets chosen for the study had associated angular/vertical bone loss with a two- or three-walled defect (Stage 3: 2018 Classification), as evaluated by transgingival probing. Only systemically healthy individuals with a plaque index of ≤20% and a Gingival bleeding score ≤25% were recruited for the study. Pan chewers, smokers, pregnant/lactating patients, patients having any systemic disease, patients with malocclusion, teeth with mobility, and grade II and III furcation involvement were excluded from the study.

### 2.2. Clinical Evaluation

The periodontal status was assessed by measuring the probing pocket depth and clinical attachment level at 6 sites (at mesiolingual, mid-lingual, distolingual, mesiobuccal, mid-buccal and distobuccal locations) around each tooth with a manual probe (University of North Carolina (UNC)-15 periodontal probe). Plaque index (O’Leary, 1972) and gingival bleeding index (GBI) (Ainamo and Bay, 1975 [14]) were noted (Table 1 and Table 2).

### 2.3. Clinical Recordings and Presurgical Procedures

Selected sites that presented with clinical attachment loss ≥3 mm and ≥5 mm probing pocket depth, associated with two wall defects as evaluated by transgingival probing, were subjected to measurements replicated by occlusal stent with groove for reproducibility of probing angulations. Complete ultrasonic scaling, root planning by hand, and mechanical instrumentation were performed, and oral hygiene instructions were given. Trans crevicular probing was performed before surgery in order to determine the bony defect, extension and morphology, probing depth, and horizontal bone loss. Treatment of carious lesion if any and anatomic factors were considered before the surgical procedure, such as the modification of over-hanging restorations and malposed teeth).

### 2.4. Surgical Procedures

Local anesthesia was given at the surgical site (2% lignocaine with 1:80,000 adrenaline). As per the principles of SFA (Figure 2), the Buccal mucoperiosteal flap was raised. A buccal envelope flap was made using Sulcular incisions in the surgical area following the gingival margin of the teeth. A horizontal or oblique butt joint incision was made at the level of the interdental papilla, and the interdental papilla was untouched. Minimal mesiodistal extension was maintained, while also ensuring adequate access debridement of the defect. The required amount of pristine supracrestal soft tissue along with the undetached oral papilla was preserved, in order to ensure flap adaptation and clear surgical access to the defect. Root debridement was performed using both mechanical and ultrasonic methods.

Upon completion of the root instrumentation, defect depth, defect width, and wall present were measured. At wound closure, in the control group (SFA alone), a vertical internal mattress suture (Vicryl 5.0, Ethicon, Sommerville, NY, USA) was placed between the base of the attached oral papilla and the buccal flap to ensure buccal flap repositioning. Meanwhile, in the test group (SFA with LLLT), the exposed bony defect was irradiated with low-level laser before flap closure. The overall energy density per irradiation was 4–5 min. A diode laser (Picosso) was used via an optical fiber with an intensity of 790–810 nm and 0.4–0.7 watts for a duration of 4–5 min for each defect (Figure 3) (Onur Ozcelik 2008) [15], using a continuous wave for Bio stimulation. During irradiation, the probe tips were placed perpendicular while in contact with both the buccal and the lingual side of the periodontal defect area.

To prevent any eye injury, both the patient and clinician wore safety goggles during laser irradiation. A vertical external mattress suture was placed at the surgical site similar to that in the control group in order to ensure repositioning. Primary flap closure was obtained by suturing (Figure 4). A coe-pack was placed at the sutured site. A single experienced operator performed all of the surgical procedures.

### 2.5. Post-Surgical Management

No mechanical oral hygiene procedures were to be performed in the surgical area for a period of 4 weeks. Chlorhexidine mouthrinse (0.12%) was given to control local plaque formation. Analgesic medications were given (ibuprofen 400 mg every 8 h) for three days. Sutures were removed after 2 weeks and the surgical site was evaluated on the basis of the Early Healing Index (Wachtel et al. 2003) [16], which consists of 5 different degrees.

Complete flap closure—no fibrin line in the interproximal area;Complete flap closure—fine fibrin line in the interproximal area;Complete flap closure—fibrin clot in the interproximal area;Incomplete flap closure—partial necrosis of the interproximal tissue;Incomplete flap closure—complete necrosis of the interproximal tissue.

### 2.6. Recall Visits

Subjects were reviewed after the 1st week, 2nd week, 3rd month and 6th month. The examiner who evaluated the clinical parameters at different time periods was blinded to the surgical intervention implemented. Clinical parameters like bleeding scores, plaque index, CAL (clinical attachment loss), and PPD (probing depth) were assessed at 3rd and 6th months after surgery. Oral hygiene was maintained by all patients during the course of therapy. During each visit, oral prophylaxis was performed.

### 2.7. Statistical Analysis

Statistical analysis was performed using SPSS 23.0 software. The comparison of parameters between the control and the test groups was performed using ANOVA and independent *t*-test.

## 3. Results

A total of 40 intrabony defects were included in the study. The descriptive parameters measured in the control and test groups are mentioned in Table 1. The plaque indexes of both of the study groups at the subsequent time points are given in Table 2. No significant differences were found in the plaque scores when comparing the groups (*p* > 0.05). There was a significant lowering of the FMBS scores in the test group compared to the control group at the 3rd and 6th months (*p* < 0.05). The present study revealed that PPD reduction was highly significant within the groups. Both groups showed a gradual reduction in probing pocket depth. When comparing the decrease in PPD between the study groups at the 3rd month, there was not much statistical significance, and at the 6th month, the test group showed a minimal increase in PPD reduction clinically compared to the control group, but this was not statistically significant. The present study revealed an increase in CAL gain in both SFA alone and SFA with LLLT from baseline to the 3rd and 6th months, with a high degree of statistical significance of *p* < 0.0001 (data are depicted in Table 2, Table 3, Table 4, Table 5 and Table 6).

## 4. Discussion

The key factors that play a major role in the clinical outcomes of periodontal surgery were classified by Kornman [17] as local site characteristics, bacterial contamination, innate wound surgical procedures, and healing potential. This study addresses all these characteristics, which have a bearing on the multi-stage process of wound healing, while also accelerating collagen production and enhancing the overall stability of the connective tissues.

Studies determining the effects of LLLT in periodontal ligament cells [18,19] have shown that LLLT can potentially stimulate the production bFGF, which helps in proliferation and differentiation of fibroblasts [20].

The present study therefore compared the clinical outcomes of SFA alone with those of SFA + LLLT in deep intrabony defects. Forty patients were treated, of whom twenty received additional LLLT for biostimulation.

Primary wound healing is of major importance in the outcome of regenerative therapy. In the current study, primary healing was evaluated using the Early Wound Healing Index, where the margin of the wound is evaluated using a five-point score. This revealed all of the patients except two showed complete closure of the flap without the formation of a fibrin line in the interproximal area at two weeks after surgery. This helps to achieve faster revascularization and facilitates primary intention healing. Those two patients in the SFA group had a score of 2, indicating complete closure of flap with the fibrin line in the interproximal area, which is in agreement with Retzepi in 2007 [21]

Subjects treated with LLLT revealed complete flap closure in all 20 of the treated sites, presenting a highly positive significant effect on wound healing, which is in agreement with the results of Bouneko et al. 2000 [22]. LLLT can accelerate wound healing, enhance bone and collagen formation, and induce anti-inflammatory effects [23]. It also induces an increased blood flow to the irradiated site by decreasing the isometric tension of vascular smooth muscle.

As a result, a higher tensile strength is achieved in the flap margins, which subsequently prevents the collapse of the healing wound, which may have been the reason for the complete wound closure in the test group.

Dental plaque is the primary etiological factor of periodontal diseases. In the current study, FMPS was calculated using the O’Leary plaque index. The results showed a significant increase in baseline to the 3rd and 6th months in both groups. No statistically significant difference was found between the study groups at any time point.

The increase in plaque score from baseline to 3 months may be due to patient apprehension due to injury to the site when brushing. This was, however, reduced after 6 months due to continued patient education and motivation. The results revealed that patients in both the groups exhibited oral hygiene at a similar level.

Gingival bleeding is considered to be an indicator of periodontal disease. In this study, FMBS was significantly reduced from baseline to 3rd months. The bleeding scores from baseline to the 3rd and 6th months decreased with high significance in the test group. When comparing the FMBS score, the bleeding sore in the test group was statistically significant at 3rd and 6th months.

While considering bleeding at the surgical site, it was 15% in the control group and 5% and10% at the surgical site of the test group. This finding is consistent with the work done by Gokce Akykol et al. in 2011 [24], who evaluated the effect of LLLT in addition to non-surgical periodontal therapy among chronic periodontitis patients with and without smoking habits. At all times, the LLLT group showed significantly greater improvement in the sulcus bleeding index (SBI). Controlled lab trials found that LLLT can cause inhibition of cyclooxygenase -2 (COX2) and reduce inflammation through the reduction of PGE2 levels in cell cultures, which could be the reason for the reduced bleeding scores in the control group.

The present study revealed that the reduction in PPD was highly significant within the groups. Both groups showed a gradual reduction in probing pocket depth. When comparing the PPD reduction between the groups at the 3rd month, there was not much statistical significance, while at 6 months the test group showed a minimal increase in PPD reduction compared to the control group, but the results were not statistically significant. These results are in accordance with those presented in studies performed by Cortellini [25] and Ashank Mishra [26].

The present study revealed an increase in CAL gain in both SFA alone and SFA with LLLT from baseline to the 3rd and 6th months, with a high significance of *p* < 0.0001. These study results are in accordance to those presented in the study performed by Trombelli [27], where it was stated that the amount of CAL gain obtained with SFA is greater than that reported for conventional flaps. The CAL gain between the study groups was revealed to be higher in the test group than in the control group at 3 months, with a significance of *p* = 0.05. These results are supported by the results presented in the studies performed by Trombelli, Roberto Farina [28,29,30]. The reasons for the increase in CAL gain could be the impact of LLLT on hard and soft tissues, resulting in a decrease in inflammatory conditions [31,32] and stimulation of bone matrix formation in osteoblasts [33], enabling periodontal tissue attachment and bone regeneration.

The buccal flap approach included in this study helps in flap suturing and repositioning. Further, it can be easily stabilized to the undetached papilla, maintaining the presurgical aesthetics and establishing a better preservation of blood supply in the interdental area.

The design of the SFA makes it possible to minimize flap elevation, which enhances wound stability and prevents the collapse of papilla into the defects. This in turn causes an increase in blood clot stability, which is a prime factor in enhancing the regenerative potential of a wound [34]. This study did not blind the evaluators as to which group the patient belonged, thereby contributing an inbuilt bias in this study.

Radiographic assessment of alveolar bone could have been performed and this is a further limitation. Despite this limitation, this study demonstrated greater PPD reduction, CAL gain and improved wound healing when compared to SFA alone.

Therefore, this study proved that the use of LLLT in combination with SFA has high potential to favorably modulate periodontal wounds in the post-surgical phase.

## 5. Conclusions

This study showed that early wound healing was uneventful. PPD reduction and increase in CAL at the 3rd and 6th months were rose gradually in both groups. The current study also confirms the positive effect of LLLT on CAL gain; therefore, LLLT can be combined with SFA to potentially enhance early wound healing and achieve higher clinical outcomes in terms of an increase in CAL gain and a decrease in PPD.

On reviewing the literature, this is the first study to assess the combining effect of LLLT alone with SFA in intrabony defect management.

## Figures and Tables

**Figure 1 healthcare-10-01301-f001:**
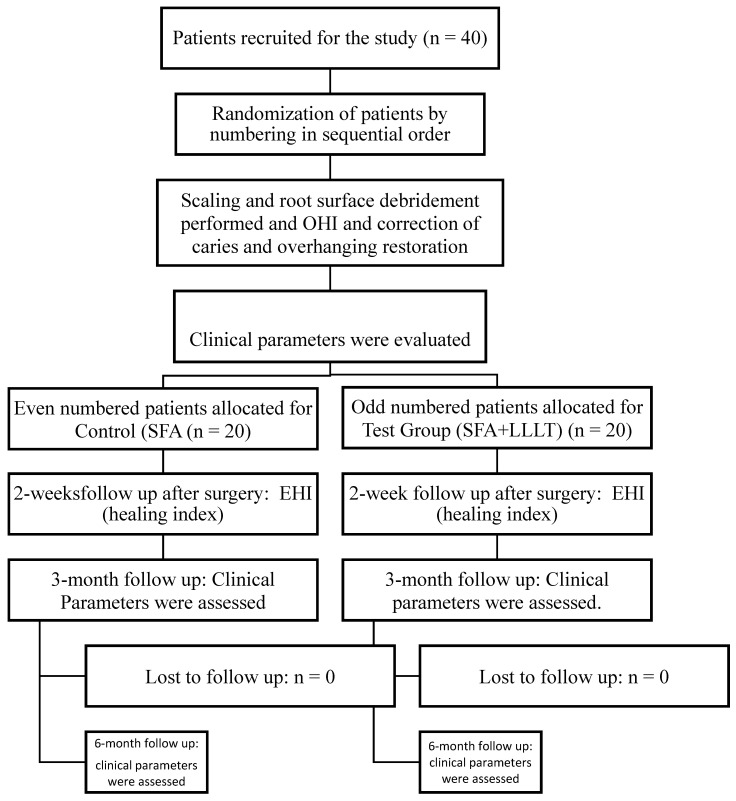
Workflow of study. Clinical parameters PPD—Probing Pocket depth, CAL—Clinical Attachment Loss, GBIGingival Bleeding Index, PlI—Plaque index.

**Figure 2 healthcare-10-01301-f002:**
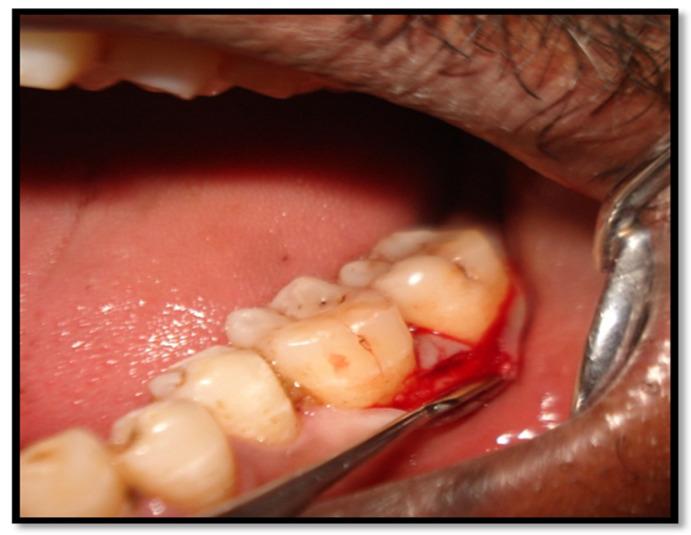
Single flap incision technique (buccal single incision placed).

**Figure 3 healthcare-10-01301-f003:**
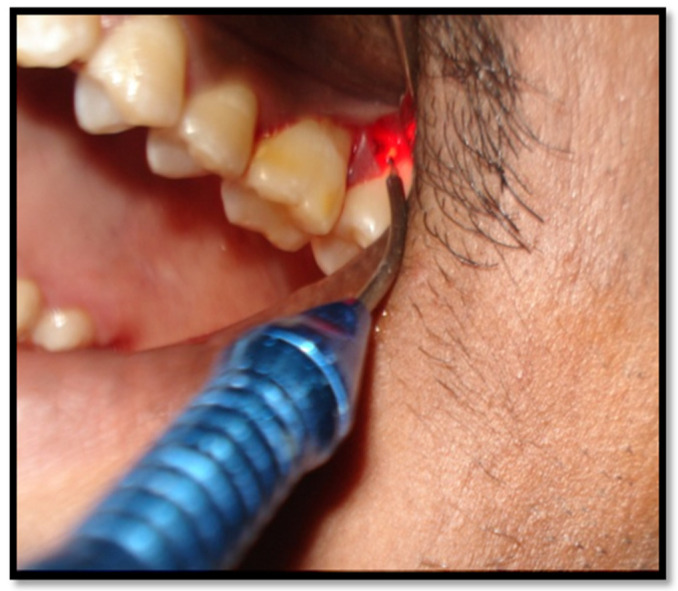
Application of low-level laser therapy on the debrided intrabony defect.

**Figure 4 healthcare-10-01301-f004:**
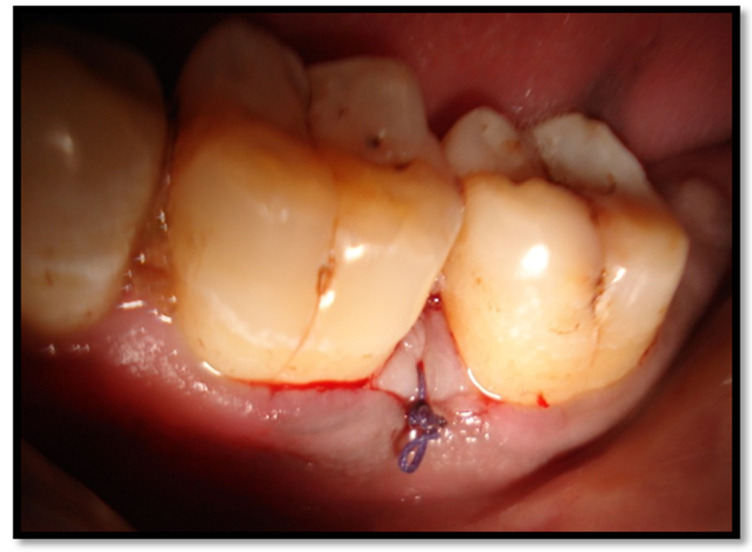
Vertical internal mattress sutures placed for flap approximation.

**Table 1 healthcare-10-01301-t001:** Descriptive statistics for test and control groups.

Parameters	Baseline (Mean ± SD)	3 Months (Mean ± SD)	6 Months (Mean ± SD)	Baseline to 3 Months (Mean ± SD)	Baseline to 6 Months (Mean ± SD)	3 to 6 Months (Mean ± SD)
FMPS(%) [Control group]	17.86 ± 1.109	25.22 ± 6.863	22.98 ± 6.030	NA	NA	NA
FMPS(%) [Test group]	18.36 ± 1.375	24.31 ± 5.229	20.42 ± 5.799	NA	NA	NA
FMBS(%) [Control]	18.19 ± 2.806	12.26 ± 3.404	10.75 ± 3.696	NA	NA	NA
FMBS(%) [TEST]	6.30 ± 1.196	3.12 ± 0.559	2.62 ± 0.406	3.25 ± 1.14	3.75 ± 0.910	0.88 ± 0.67
PPD(mm) [Control]	6.30 ± 1.196	3.12 ± 0.559	2.62 ± 0.406	3.25 ± 1.14	3.75 ± 0.910	0.88 ± 0.67
PPD(mm) [Test]	6.68 ± 0.963	3.08 ± 0.684	2.38 ± 0.582	3.60 ± 1.14	4.30 ± 0.95	0.70 ± 0.52
CAL(mm) [Control]	3.58 ± 1.042	1.48 ± 0.819	0.88 ± 0.626	2.10 ± 1.05	2.70 ± 1.36	0.60 ± 0.74
CAL(mm) [Test]	4.25 ± 1.323	1.32 ± 1.270	0.80 ± 1.342	2.93 ± 1.54	3.45 ± 1.22	0.53 ± 0.87

**Table 2 healthcare-10-01301-t002:** Independent *t*-test for plaque score.

*t*-Test for Equality of Means						
Comparing FMPS Scores between Control and Test						
FMPS	t	Df	Sig. (2-Tailed)	Mean Difference	95% Confidence Interval of the Difference	
					Lower	Upper
Comparing control and test at baseline	−1.258	38	0.216 (NS)	−0.497	−1.297	0.303
Comparing control and test at 3 months	0.472	38	0.639 (NS)	0.912	−2.994	4.817
Comparing control and test at 6 months	1.370	38	0.179 (NS)	2.563	−1.224	6.349

**Table 3 healthcare-10-01301-t003:** Independent *t*-test for bleeding scores between Control and Test groups.

FMBS	t	Df	Sig(2 Tailed)	Mean Difference	95% Confidence Interval of the Difference	
					Lower	Upper
Comparing control and test at baseline	2.166	38	0.037(sig)	4.417	0.289	8.545
Comparing control and test at 3 months	2.260	38	0.030(sig)	3.630	0.378	6.882
Comparing control and test at 6 months	2.710	38	0.010(sig)	5.749	1.454	10.043

**Table 4 healthcare-10-01301-t004:** Independent *t*-test for the difference in PPD reduction.

Variable	Group	N	Mean	Std. Deviation	*t*-Test	*p*-Value
Baseline to 3 months	Control	20	3.25	1.14	−0.969	0.34(NS)
	Test	20	3.60	1.14		
Baseline to 6 months	Control	20	3.75	0.91	−1.868	0.07(sig)
	Test	20	4.30	0.95		
3 to 6 months	Control	20	0.88	0.67	0.924	0.36(NS)
	Test	20	0.70	0.52		

**Table 5 healthcare-10-01301-t005:** Independent *t*-test for the difference in mean CAL gain.

Variable	Group	N	Mean	Std. Deviation	*t*-Test	*p*-Value
Baseline to 3 months	Control	20	2.10	1.05	−1.981	0.05(sig)
	Test	20	2.93	1.54		
Baseline to 6 months	Control	20	2.70	1.36	−1.833	0.07(NS)
	Test	20	3.45	1.22		
3 to 6 months	Control	20	0.60	0.74	0.295	0.77(NS)
	Test	20	0.53	0.87		

**Table 6 healthcare-10-01301-t006:** Independent *t*-test for EHI comparing Test and Control.

EHI	**t-Test for Equality of Means**					
	**t**	**Df**	**Sig. (2-tailed)**	**Mean Difference**	**95% Confidence Interval of the Difference**	
	1.453	38	0.154(NS)	0.100	lower	Upper
					−0.039	0.239

## Data Availability

Not applicable.

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
