# Peer review of "The Efficacy of Low-Level Laser Therapy Combined with Single Flap Periodontal Surgery in the Management of Intrabony Periodontal Defects: A Randomized Controlled Trial"

_healthcare, 2022, doi:10.3390/healthcare10071301_

Round 1
Reviewer 1 Report
Overall, the manuscript is scientifically written with proper results and discussion.
Author Response
S.NO |
REVIEWER-1 COMMENTS |
CORRECTIONS DONE |
1 |
NIL |
- |
|
|
|
|
REVIEWER -2 |
|
1 |
TITLE CHANGE |
Page:1,Line :2,3,4CORRECTED |
2 |
ABSTRACT--SINGLE PARAGRAPH -PRE REGISTRATION NUMBER -ANIMAL STUDIES
-to assess
Not to include the number of defects LESS STRONGER CONCLUSION: -Indicate positive effects -LLLT+SFA can be combined .
|
CORRECTED Page3:Line 113ETHICAL APPROVAL NUMBER GIVEN
Page:1 Line 92-94
Page:1,Line35Assessing(CORRECTED)
CORRECTED.
Page:1,Line 45Suggested positive effects(CORRECTED) Page:1,Line 45may be combined(CORRECTED) |
3 |
INTRODUCTION:
Definition, epidemiology, common approaches and treatments should be reported
connection between periodontitis and systemic diseases,
TMJ and Periodontitis
LLLT –Low Level Laser Therapy
Line 96-which LLLT used
No of defects in result section
Wound healing Reference
|
Page2: 51-61CORRECTED
CITATIONS INCLUDED
CITATIONS INCLUDED
Page 2:Line 70CORRECTED
Page 3:Line 116{DIODE LASER}
Page 7:Line249(CORRECTED)
Page2:Line 68Reference added
|
4 |
MATERIALS AND METHODS:
Interventional study
No of cases |
Page3:Line 109RCT(Corrected)
Page3:Line115.40 cases (CORRECTED)
|
1. |
REGISTRATION NUMBER
Declaration of Helsiniki
CONSORT GUIDELINES
SAMPLE SIZE CALCULATION
REVIEWER:3
INTRODUCTION:
Line 71
LINE 80-81 AUTHORS
SAMPLE SIZE CALCULATION
AGE 25 -50?
2018 CLASSIFICATION
LINE 155-156
RECESSION
X-RAYS |
Page3:Line 113.number given
Page 3:Line114added
Page 6: 215,216 Page 7: 233-235
ADDED
Page7:Line 234-235.Expanded(CORRECTED) Page3:Line127Included(CORRECTED)
Page2:Line 73Particular wavelength(CORRECTED)
Recently:Page2 :Line 85 removed(CORRECTED)
LINE REMOVED
Page3:Line127Included(CORRECTED) INCLUDED(CORRECTED) Because intrabony defects are primarily involved in this age group
Page3:Line121
Page5:Line 172.LINGUAL(CORRECTED)
Recession not found in the cases
Page7:Line 243INCLUDED |
Academic editor comments:
Dear Authors please verify if what suggested by referees has been introduced/modify and also check references (I.E. n 28):
Reply: As advised, reference 28 was verified. Referees comments are addressed

Reviewer 2 Report
Dear Authors,
this study aimed at evaluating if the clinically observed outcomes following periodontal surgery (single flap approach) could be favourably enhanced by the adjunctive use of Low-Level Laser Therapy (LLLT). Moreover, this randomised controlled trial aimed at observing if the combining effect of single flap approach surgery and LLLT would possibly alter and enhance the biology of periodontal wound healing.
The study is of scientific interest and in line with the aims of the journal, but the methodology should be improved and the manuscript should be copyedited by a native English speaker or copyediting service.
Title:
“Low-Level Laser Therapy in the management of intrabony periodontal defects using single flap approach -Randomized controlled trial”. In order to highlight the RCT study design, I suggest to modify the title as follow: “The efficacy of Low-Level Laser Therapy combined to single flap periodontal surgery in the management of intrabony periodontal defects: a randomized controlled trial”.
Abstract
- The abstract should be a total of about 200 words maximum. The abstract should be a single paragraph and should follow the style of structured abstracts, but without headings: 1) Background: Place the question addressed in a broad context and highlight the purpose of the study; 2) Methods: Describe briefly the main methods or treatments applied. Include any relevant preregistration numbers, and species and strains of any animals used. 3) Results: Summarize the article's main findings; and 4) Conclusion: Indicate the main conclusions or interpretations. The abstract should be an objective representation of the article: it must not contain results which are not presented and substantiated in the main text and should not exaggerate the main conclusions. (https://www.mdpi.com/journal/ijerph/instructions).
- Please change “The study aimed to assess the clinical outcomes” to “The study aimed at assessing the clinical outcomes”.
- In the Methodology section, you have not to report the number of the included defects. Please add these information in the results section.
- “Conclusion: These data indicate the positive effect of LLLT over CAL gain, thus LLLT can be combined with SFA to potentially enhance the early wound healing and higher clinical outcomes in terms of increase in CAL and decrease in PPD”. Please, report less stronger conclusion, for example: “The results of the present RCT suggested the positive effect of LLLT over CAL gain. LLLT may be combined with SFA to potentially enhance the early wound healing and higher clinical outcomes in terms of improvement of CAL and PPD.”
Introduction
- The Introduction section should be better reported. Definition, epidemiology, common approaches and treatments should be reported. Moreover, explain the connection between periodontitis and systemic diseases, explaining how periodontitis can negatively affect the general state of health. In literature, it is supposed the dual interaction between periodontal disease and systemic disease (please cite: doi: 10.3390/jcm10194578, doi: 10.3233/NRE-201537).
- I suggest to report that LLLT is a valid approach in other fields of dentistry, as temporomandibular disorders (please cite doi: 10.3233/BMR-210236, doi: 10.1016/j.bjoms.2022.03.013, doi: 10.1007/s10103-021-03467-y.).
- When you cite for the first time LLLT you have to report “Low-Level Laser Therapy (LLLT)” (also in the text, not only in the Abstract section”.
- Line 96. You have to report which LLLT you used.
- Please the number of the included defects/ patients must be reported in the Results section.
- Numerous advancements are being made in understanding the biology of wound healing in recent years. Wound healing is characterized by proliferation, differentiation migration, and adhesion of different types of cell. Add references.
- All of these activities are triggered when different growth factors regulate different cell functions like TGFβ and VEGF help in angiogenesis, connective tissue growth factor and bFGF help in fibroblast and collagen synthesis. These growth factors play a prime role ithe n healing of periodontal tissue. Add references.
Materials and Methods:
- “This is an interventional study”. This sentence should be reformulated reporting the type of study design (RCT).
- How many patients were included in the study?
- Was this study registered to ClinicalTrials.gov or others? If not, you should do it. Please report the number of registration.
- Was the study performed according to the Declaration of Helsinki?
- The protocol should follow the guidance from the CONSORT Guidelines (Moher D, Hopewell S, Schulz KF, et al. CONSORT 2010 explanation and elaboration: updated guidelines for reporting parallel group randomized trials. Int J Surg. 2012;10(1):28-55.). Please change Figure 1, accordingly. Moreover, you have to report all the abbreviations (e.g. CAL, PPD, GBI, ….).
- Did you calculate the sample size?
Results
In all Tables you have to report all the abbreviations.
Discussion
The discussion section was well described.
The topic is very interesting and actual. The study is of scientific interest and in line with the aims of the journal, but the methodology should be improved and the manuscript should be copyedited by a native English speaker or copyediting service.
Author Response

(The authors gave the same response as above.)

Reviewer 3 Report
The present study aimed to investigate the use of low laser therapy in the management of intrabony defects.
While the argument is timely and clinically relevant, the manuscript lack’s structure, methodologically inaccurate and statistically wrong. Moreover, the manuscript requires extensive proof-reading, some sentences are wordy and odd. The authors
might consider addressing the following points, as listed below, to enrich their report prior to future submissions.
Introduction
Which is the novelty that this paper brings? Advantages and disadvantages of previous protocols.
The introduction is poor of arguments and not in line with the aim of the study.
e.g., line 60 changed by irradiation at a particular …. Missing information’s
line 71 recently (2007) we are in 2022 is not anymore recent procedure.
Line 80-81 seems a speculation of the authors no evidence on it.
Materials and method
Was the protocol registered on a trial’s registration platform?
This is crucial for the reviewers who can check all the methodological issues.
Was a sample size calculated?
Without a sample size all the data extracted are inaccurate and probably not statistically relevant.
Figure 1 needs some changes is not easy to understand.
The description of intrabony defect only with a probe seems a limitation of the study moreover why only patients ranged between 25-50 years old?
What type of periodontal patients were involved if infrabony defect are presents for sure they have periodontist (classification 2018) stage and grade is missing.
The periodontal status was only assessed with probing depth where are the x-rays crucial for the evaluation of the overall bone loss and description of the defects?
Surgical procedure
Single flap approach was introduced as an envelope flap only on one aspect of the defect it could be buccal or lingula according to the extension of the defect please correct the sentence
(line155-156)
Why were the sutures removed after 2 weeks?
The biological principles described the wound healing in the range of a week.
How was possible to measure CAL without mention the recession?
Please also provide data regarding REC.
According to the result there is not statistically significant difference between the use or not of low laser therapy. Moreover, without the clinical assessment of REC and the use of x-rays the study seems limited and not completed.
Author Response

(The authors gave the same response as above.)

Round 2
Reviewer 2 Report
In this study, the authors aimed at evaluating if the clinically observed outcomes following periodontal surgery (single flap approach) could be favourably enhanced by the adjunctive use of Low-Level Laser Therapy (LLLT). Moreover, this randomised controlled trial aimed at observing if the combining effect of single flap approach surgery and LLLT would possibly alter and enhance the biology of periodontal wound healing.
The study was improved. It is suitable for publication.
Reviewer 3 Report
Dear authors,
thank you for providing the modified version of your manuscript